# Relative Efficiency of Pitfall vs. Bait Trapping for Capturing Taxonomic and Functional Diversities of Ant Assemblages in Temperate Heathlands

**DOI:** 10.3390/insects12040307

**Published:** 2021-03-30

**Authors:** Axel Hacala, Clément Gouraud, Wouter Dekoninck, Julien Pétillon

**Affiliations:** 1UMR CNRS 6553 Ecobio, Université de Rennes, 263 Avenue du Gal Leclerc, CS 74205, CEDEX, 35042 Rennes, France; clementgouraud@hotmail.fr (C.G.); julien.petillon@univ-rennes1.fr (J.P.); 2EA Géoarchitecture: Territoires, Urbanisation, Biodiversité, Environnement, Université de Bretagne Occidentale, CS 93837, CEDEX 3, 29238 Brest, France; 3Royal Belgian Institute of Natural Sciences, Vautierstraat 29, 1000 Brussels, Belgium; wdekoninck@naturalsciences.be

**Keywords:** sampling method, estimated richness, functional diversity, maritime cliffs, Western France, Formicidae

## Abstract

**Simple Summary:**

Ants, due to their high ecological diversity, are challenging to properly sample. This issue has been addressed by many authors devising multiple sampling techniques. Depending on the habitats sampled, the effectiveness and complementarity of the sampling techniques may vary. Only little work has been done in open temperate habitats. This study aimed to assess the relative efficiency of two common sampling methods: pitfall and bait trapping. The comparison was performed using both a taxonomic (species count) and a functional (i.e., acknowledging of ecological characteristics of species to describe an assemblage of species) approach. Pitfall traps captured more species and a wider set of functional traits than did bait traps, and all species caught by bait traps were also caught by pitfall traps. It therefore appears that in the particular context of open temperate habitats, using bait traps on top of pitfall traps will cost time without information gained and that pitfalls should thus be favored in this context.

**Abstract:**

Whereas bait and pitfall trappings are two of the most commonly used techniques for sampling ant assemblages, they have not been properly compared in temperate open habitats. In this study, taking advantage of a large-scale project of heathland restoration (three sites along the French Atlantic Coast forming a north-south gradient), we evaluated the relative efficiency of these two methods for assessing both taxonomic and functional diversities of ants. Ants were collected and identified to species level, and six traits related to morphology, behavior (diet, dispersal and maximum foraging distance), and social life (colony size and dominance type) were attributed to all 23 species. Both observed and estimated species richness were significantly higher in pitfalls compared to spatially pair-matched bait traps. Functional richness followed the same pattern, with consistent results for both community weighted mean (CWM) and Rao’s quadratic entropy. Taxonomic and functional diversities from pitfall assemblages increased from north to south locations, following a pattern frequently reported at larger spatial scales. Bait trapping can hardly be considered a complementary method to pitfall trapping for sampling ants in open temperate habitats, as it appears basically redundant with the latter sampling method, at least in coastal heathlands of the East-Atlantic coast.

## 1. Introduction

Because of their high abundance and diversity except in polar regions, ants play a key role in ecosystem functioning in many terrestrial habitats, from open ecosystems like deserts to forests, and from the floor to the canopy [1]. Ants are known to be good bioindicators [2,3,4,5], with high ecological importance [1]. As many other groups, ants are globally more diverse in the tropics than in temperate areas [6], and the amount of publications regarding sampling methods is also greater for tropical regions [7,8]. Temperate open habitats on the other hand have not been reviewed as much and are linked to methodological limitations regarding what kind of traps can be set in them. 

Two common sampling methods for ant assemblages are pitfall and bait traps. They were selected in this study because they do not need arboreal stratum nor extensive litter depth to be used [9]. Pitfall traps are pits set in the ground that sample epigeic fauna that randomly falls inside it. This sampling method provides good sampling effort for epigaeic fauna, yet it is sensitive to species size and level of activity [10,11,12]. On the other hand, baiting traps use foods to attract ants, which can later be actively sampled. This sampling method may attract species from different strata and can assess competition relationships between ants while being sensitive to exclusion, missed diet, and time of sampling [13]. 

Pitfall trapping is expansively used and recognized as a good method to sample epigeic arthropods [14], including ants [15]. However, it is also criticized for not being an exhaustive technique [10], suffering from several biases related to microhabitat complexity and trap diameter, reported for decades now, e.g., [16,17]. The limits of pitfall trapping are particularly obvious for ants due to their heterogeneous use of space [8] and pheromone distress signals that can induce artificially high abundance in single traps [18]. 

On the other hand, bait trapping is recognized as the most common method for sampling ants [13] and is sometimes used simultaneously with pitfall trapping, but mostly in tropical habitats [19]. A few examples can be found for temperate areas, but they are restricted to closed habitats like forests [20,21,22,23]. 

In this study, we compared the efficiency of pitfall vs. bait trapping for assessing the taxonomic and functional diversities of ant assemblages in temperate open habitats (coastal heathlands). We used the sampling design provided by a large-scale project where heathland restoration is evaluated. Arthropods were sampled in three heathlands along the French Atlantic Coast covering a gradient of restoration time that we did not test here because of its small spatial scale. We especially tested the hypothesis that in an open habitat, pitfall trapping performs well see, e.g., [24] and is consequently expected to capture ant diversities in greater proportions than bait trapping does in closed habitats [21,22]. Conversely, we expected functional diversity to be inferior with bait trapping, as only some species are targeted by baits [4]. Lastly, we expected taxonomic, but not functional, diversities to differ between sites, with species richness increasing from north to south [25] because even if the gradient is geographically short, our southern location is known to have a warmer microclimate [26].

## 2. Material and Methods

### 2.1. Study Sites

Fieldwork was done at three coastal sites in Brittany, Western France. Sites were coded according to the north-south gradient they follow. La Pointe de Pen-Hir (S1), located on the mainland (48°15′03″ N, 4°37′25″ W), La Pointe de l’Enfer (S2) (47°37′18.3″ N 3°27′46.9″ W), and L’Apothicairerie (S3) (47°21′44.0″ N, 3°15′34.9″ W) (see [27], 2020 for a full description and pictures of the sampling sites). Mean annual temperatures are 10.6 °C, 12.6 °C, and 12.6 °C, mean maximal temperatures are 13.13 °C, 15.08 °C, and 14.86 °C, and mean annual precipitations 537.8 mm, 717.0 mm, and 675.6 mm, for S1, S2, and S3 respectively [28]:data from association infoclimat.fr.

### 2.2. Sampling Design

Two 400 m^2^ plots of homogeneous vegetation were designed for each of the three degradation states at each of the three sites, resulting in six plots per site. Four pitfall traps (80 mm in diameters and 100 mm deep) were set at each plot. Traps were half-filled of salted solution (250 g/L) with a drop of odorless soap, and settled 10 m apart in order to avoid interference and local pseudoreplication [29]. This resulted in a total of 71 traps (Figure 1) (in one station, the sampling area was too restricted to set four traps spaced 10 m apart, so one was removed) active between mid-March and mid-June 2017, and emptied every 2 weeks. One bait trap was spatially pair-matched with each pitfall trap, resulting in 71 bait traps. The baiting device consisted of a cardboard square (4 cm × 4 cm) on which approximately 1 cm^3^ of tuna rillettes and a few drops of honey were deposited (Figure 2). A wooden stick was driven through the cardboard, which anchored it to the ground to ease both sampling and detection on the field. Bait traps were set five times for 2 h in the middle of the day and by sunny weather only, 2 weeks apart between March and June. All the ants present on the trap after the 2 h span were captured for identification. The pitfall traps remained active for 12 weeks from mid-March to mid-June. For each trap, data from the whole season were pooled. This pooling was performed in order to assess local assemblages regardless of potential phenological variations.

Samples of pitfall and bait traps were sorted, transferred to ethanol 70°, and stored at the University of Rennes 1. Ants were identified to species level using keys of Blatrix [30] and of Seifer [31,32,33]. For further sites description see [27].

### 2.3. Functional Traits

Six traits related to morphology, behavior (diet and dispersal), and social life (colony size and dominance type) were attributed to all the 23 species (Appendix A), using different bibliographic sources (Appendix B).

### 2.4. Statistical Analysis

Presence/absence data were used to avoid abundance bias from difference in species activity rate and/or in sensitivity to habitat structure [34]. Species richness was calculated with the vegan package [35], while functional richness as well as Rao’s quadratic entropy and the CWM (community-level weighted means) were calculated with the FD package [36]. Species richness was also compared between methods using estimated richness based on methods developed by Chao [37,38] using the “iNEXT” function in the iNEXT package [39]. This method was selected to account for the possible influence of sampling coverage. The test was ran with 40 knots and 200 bootstrap replications. Significant differences were further assessed using the absence of overlapping confidence intervals on iNEXT curves [39,40].

The influence of sampling methods on species richness was tested using a Poisson GLMM (generalized linear mixed model), while functional richness and Rao’s quadratic entropy used a Gaussian GLMM; all three used site as a fixed factor, as ants diversity is known to increase in warmer climates [41]. The GLMM type of error (Poisson vs. quasi-Poisson) was assessed following O’Hara & Kotze [42] for the Poisson model. If detected, overdispersion was handled by using a quasi-Poisson distribution. Full models were tested and underwent decremental fitting. Functional patterns were compared between sampling methods with CWM and Rao’s quadratic entropy to assess shifts in main trait values and trait divergences, respectively [43]. The CWM based on numerical attributes (colony size and foraging distance) was tested using a Wilcoxon test.

All analyses were carried out using R software (version 3.6.1 2019-07-05).

## 3. Results

Pitfall and bait trapping resulted in the collection of 4976 and 4419 individuals, respectively (Appendix A), altogether representing 23 species. All species were collected by pitfall traps, and 10 by bait traps (Table 1), resulting in 13 species caught in pitfall but missed by bait. *Formica pratensis* was sampled by bait traps only in S1 and S2, and it was sampled with both methods in S3. Sampling coverage reached asymptotes (Figure 3) for both sampling methods and was above 90%, indicating a sufficient sampling intensity. The observed species richness was significantly higher in pitfalls than in bait traps (χ^2^_1,130_ = 0.74; *p* < 0.001) (Figure 4a).

The estimated species richness was significantly higher in pitfall than in bait traps when plotted against the number of samples or vs. the sampling coverage (Figure 3b,c, respectively).

The same pattern was observed for functional richness, with higher functional richness in pitfall compared to bait traps (χ^2^_1,130_ = 0.71; *p* < 0.001) (Figure 3b). The CWM displayed the same main trait value with the same categorical variable (e.g., large dominant omnivorous epigeic ants with independent colony formation), while no significant differences were observed in the colony size score (W = 207; df = 2; *p* = 0.546). The foraging distance and Rao’s quadratic entropy were both significantly higher in pitfall than in bait traps (W = 2880.5; df = 2; *p* = 0.002 and (χ^2^_1,130_ = 0.17; *p* < 0.001, respectively; see Figure 4c).

For both methods combined, significant differences were observed along the north-south gradient in both taxonomic and functional diversity metrics (Figure 5), with higher species richness in the south (χ^2^ = 6.20; df = 2; *p* = 0.045), lower functional richness in the north (χ^2^ = 8.21; df = 2; *p* = 0.01), and higher Rao’s quadratic entropy in the south (χ^2^ = 11.12; df = 2; *p* = 0.004).

## 4. Discussion

Following our first assumption, both observed and estimated species richness were higher in pitfall compared to bait traps, with bait traps capturing only a subset of what the pitfall traps did. This results concur with the literature concerning closed habitat ant assemblages [21,22], with bait traps used with other sampling methods being outperformed. Several hypotheses can be formulated to explain this result. One could argue that the sampling effort greatly differed between the two methods, with pitfall traps being active for 2 months in a row and bait traps for a total of 10 h covering five events of sampling with a 2 h span each. Although the sampling time (2 h) of bait traps could be a reason for its lower effectiveness, the sampling coverage was very high for both methods, and their capture ability was thus comparable regardless of the sampling effort. Previous studies with higher sampling effort with baiting also showed similar trends [23]. The sampling time might still have an effect, since circadian activity is known to vary greatly between ant species [44,45]. Species being active at night, early day, or late afternoon could therefore have been missed by bait trapping with the protocol used here. Another known bias known of bait traps is exclusive competition [46]. Some competitive ants could have monopolized the baits and limited the access to other less competitive ants. Such competition could explain both a high coverage and the species missed from the local pool. 

The functional analysis showed that the two methods caught mainly dominant ants, and that ant assemblages caught by bait traps did not differ significantly from the mean trait. This problem could have been avoided by a higher bait trap duration since less competitive species tend to be active during cooler times of the day to avoid interspecific competition [13]. Multiplying the number of observations during the 2 h span bait traps were active could have lessen the potential impact of exclusive competition [13]. 

Another fact can also help explaining the redundancy of the two sampling methods: species captured by bait traps are considered particularly populous and active [4]. This is known to increase the probability of capturing ants in pitfall traps [13], which together with the CWM results might explain the high similarity between the two sampling methods. 

On the other hand, several species missed by bait traps have traits that can explain their absence. Slow-moving species (i.e., *Aphaenogaster subterranea, Myrmecina graminicola, Solenopsis fugax*) have a lower probability to contact the baits, as suggested by the shorter foraging distance observed in ant species sampled in bait traps. Some species can also be absent because of their specialized diet, such as aphids’ honeydew (*Lasius flavus, Lasius emarginatus*) or seeds (*Messor capitatus*) [4,13]. These species with traits differing from the CWM are likely responsible for the higher Rao’s quadratic entropy and functional richness in pitfall traps. Lastly, some species exclusive of pitfall traps (e.g., *Aphaenogaster gibbosa, Hypoponera eduardi,*
*Plagiolepis pallescens, Ponera coarctata,* or *Tetramorium atratulum*) are considered uncommon or rare [30], which might lower the probability to contact them using baits traps, eventually contributing to the observed pattern. Another final explanation comes from the fact that the important degradation of our study sites may have increased the relative abundance of common and numerous species that might have monopolized baits [13]. 

Our last assumption for the north-south gradient was validated, with species richness higher in the southern location. This result is consistent with patterns of ant diversity often reported at larger spatial scales, e.g., [41]. More surprisingly, even at relatively small spatial scales, this N-to-S difference also applied to functional metrics, as it was found a few times at larger spatial scales, e.g., in Europe: [47]. This phenomenon could be explained by the zoogeographical area that drives diversity on the French territory, as our southern sites are located in the northern limits of a richer Mediterranean zoogeographic zonation [48].

Our conclusion, applicable in open temperate habitats such as coastal heathlands, is to avoid bait trapping for ant surveying, as it is redundant with pitfall trapping. This claim is further stressed because the results were stable for all metrics of diversity considered. Furthermore, multiplying sampling methods is costly both in terms of resources and time and should therefore be considered only when data are obviously improved. Similar conclusions were drawn by Mahon et al. [49], who stated that using several sampling methods at once was not always necessary in temperate environments, especially in studies not aiming at full inventory of ant diversity. While bait trap does not complement pitfall traps, active collection is reported to be an interesting option to complete the species inventory by pitfalls traps [13]; however, this has to be performed by individuals with highly specialized skills in order to be effective [8].

## Figures and Tables

**Figure 1 insects-12-00307-f001:**
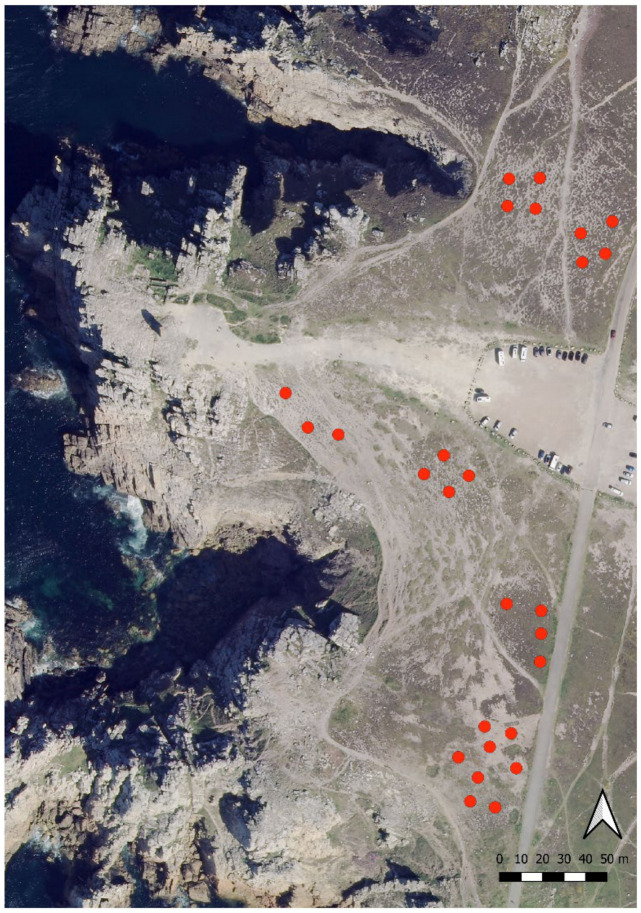
Pitfall traps arrangement in La Pointe de Pen-Hir (S1) as an example.

**Figure 2 insects-12-00307-f002:**
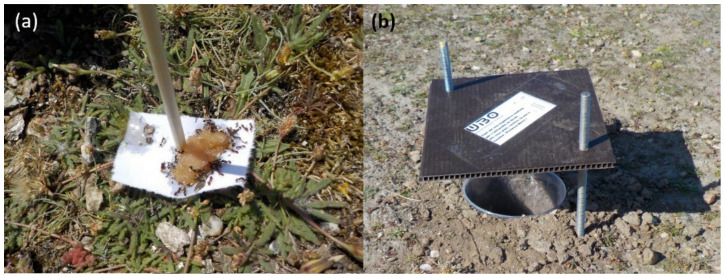
Example of (**a**) bait trap and (**b**) pitfall traps.

**Figure 3 insects-12-00307-f003:**
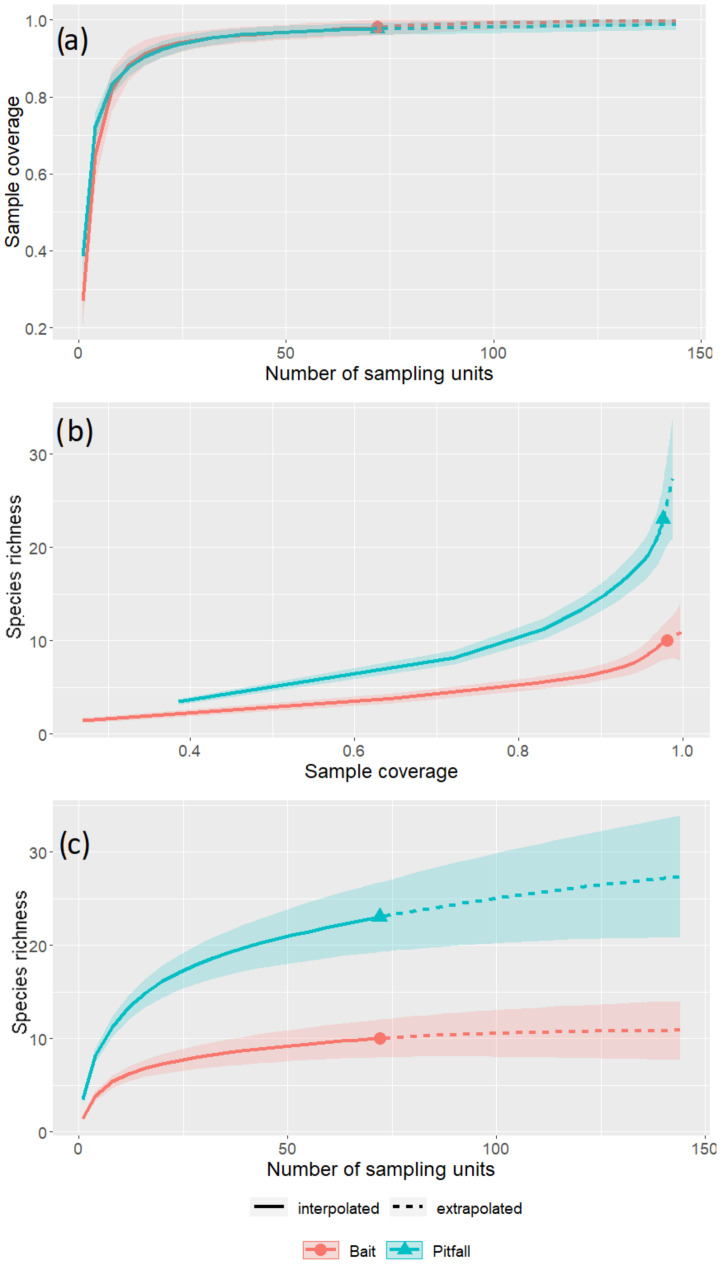
Estimated richness of ant assemblages from bait (red) and pitfall traps (blue) from the pooled data of the three sites. (**a**) Coverage vs. number of sampling units; (**b**) Species diversity vs. sampling coverage; (**c**) species diversity vs. number of sampling units. Plain line corresponds to observed data, while dashed line stands for extrapolated estimation. The colored area around the line is the standard deviation resulting from bootstrapping.

**Figure 4 insects-12-00307-f004:**
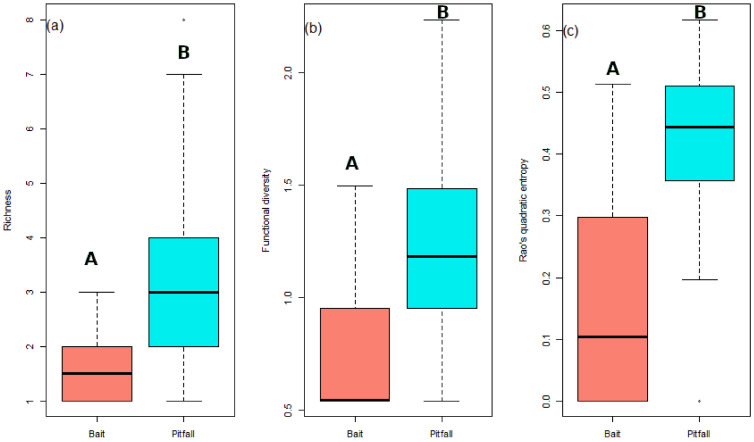
Comparison between bait and pitfall traps of (**a**) species richness; (**b**) functional diversity; and (**c**) Rao’s quadratic entropy. Significant differences are represented using different successive letters (e.g., A & B) for *p*-value < 0.001.

**Figure 5 insects-12-00307-f005:**
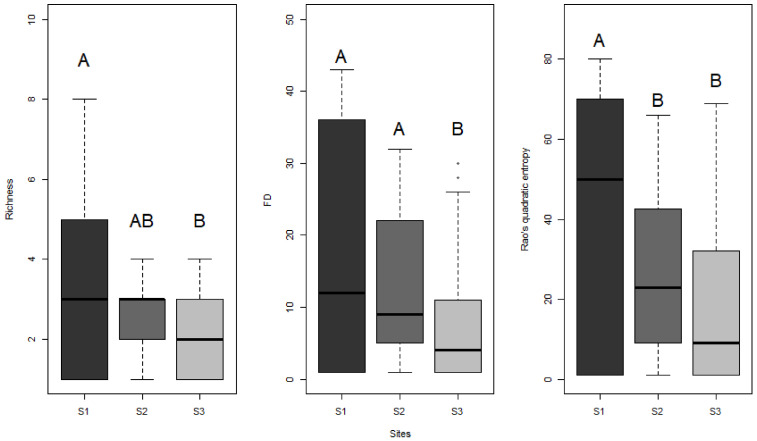
Boxplot of diversity metrics (species richness, functional richness (FD), and Rao’s quadratic entropy) compared between the three sites along a north-south gradient (S1 being in the south and S3 in the north). Significant differences (*p* < 0.05) are represented using different successive letters (e.g., A & B).

**Table 1 insects-12-00307-t001:** Occurrence of ant species in the three sites (S1; S2; S3) for both bait and pitfall traps.

Species	S1		S2		S3		All Sites	
	Bait	Pitfall	Bait	Pitfall	Bait	Pitfall	Bait	Pitfall
*Aphaenogaster gibbosa* (Latreille, 1798)	0	0	0	1	0	0	0	1
*Aphaenogaster subterranea* (Latreille, 1798)	0	0	0	0	0	1	0	1
*Formica cunicularia* Latreille, 1798	1	1	1	1	1	1	1	1
*Formica pratensis* Retzius, 1783	1	0	1	0	1	1	1	1
*Hypoponera eduardi* (Forel, 1894)	0	0	0	1	0	1	0	1
*Lasius alienus* (Foerster, 1850)	1	1	0	1	0	1	1	1
*Lasius emarginatus* (Olivier, 1792)	0	0	0	1	0	0	0	1
*Lasius flavus* (Fabricius, 1782)	0	0	0	0	0	1	0	1
*Lasius niger* (Linnaeus, 1758)	1	1	0	0	1	1	1	1
*Lasius platythorax* Seifert, 1991	0	0	1	1	0	0	1	1
*Lasius psammophilus* Seifert, 1992	1	1	0	0	0	0	1	1
*Messor capitatus* (Latreille, 1798)	0	0	0	1	0	0	0	1
*Myrmecina graminicola* (Latreille, 1802)	0	1	0	1	0	1	0	1
*Myrmica ruginodis* Nylander, 1846	0	0	0	0	0	1	0	1
*Myrmica sabuleti* Meinert, 1860	0	1	1	1	1	1	1	1
*Myrmica scabrinodis* Nylander, 1846	0	0	0	1	1	1	1	1
*Plagiolepis pallescens* Forel, 1894	0	1	0	0	0	0	0	1
*Ponera coarctata* (Latreille, 1802)	0	1	0	0	0	0	0	1
*Solenopsis fugax* (Latreille, 1798)	0	1	0	1	0	1	0	1
*Tapinoma erraticum* (Latreille, 1798)	1	1	1	1	1	1	1	1
*Temnothorax unifasciatus* (Latreille, 1798)	0	0	0	0	0	1	0	1
*Tetramorium atratulum* (Schenck, 1852)	0	0	0	0	0	1	0	1
*Tetramorium gr. caespitum-impurum*	1	1	1	1	1	1	1	1
Only bait/shared/only pitfall	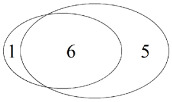		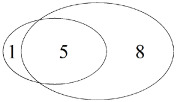		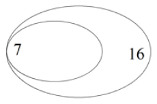		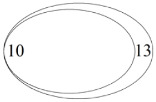	

## Data Availability

The data presented in this study are openly available in Zenodo at https://doi.org/10.5281/zenodo.4640190, reference number c63c6a2cc52c1832dd4a4afe48752097.

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
