# Peer review of "Relative Efficiency of Pitfall vs. Bait Trapping for Capturing Taxonomic and Functional Diversities of Ant Assemblages in Temperate Heathlands"

_insects, 2021, doi:10.3390/insects12040307_

Round 1

Reviewer 1 Report

The manuscript “Relative efficiency of pitfall vs. bait trapping for capturing taxonomic and functional diversities of ant assemblages in temperate heathlands” aims to evaluate three hypotheses:

  1. Pitfall trapping performs well and is consequently expected to capture ant diversities in similar proportions than bait trapping.
  2. Functional diversity is inferior with bait trapping as only some species are targeted by baits.
  3. Taxonomic, but not functional, diversities differ among sites, with species richness increasing from North to South.

This is an interesting study but Methods and Discussion require more detail and work. Despite similar patterns have been observed in other studies, there are several issues that deserve attention and discussion. The use of mathematical tools (iNEXT package) aimed to sort out possible differences in sampling coverage is correct, but the sampling design with such huge differences in sampling effort between treatments is the main flaw of this study.

Sampling design. Please write explicitly how many traps of both types were used. Also, in this section it appears as if they collected several samples per trap but this information is missing. In the results it appears (or one has to guess) as if the number of samples is similar to the number of traps. So, did they polled all samples for each trap? Why? Or how did authors statistically treat the repeated measurements? This part needs to be much better explained, otherwise the results lack of support.

Authors should also cite for the iNEXT package: Chao, A., Ma, K. H., and Hsieh, T. C. (2016) iNEXT (iNterpolation and EXTrapolation) Online: Software for Interpolation and Extrapolation of Species Diversity. Program and User’s Guide published at http://chao.stat.nthu.edu.tw/wordpress/software_download/.

L20. Traps

L83-87. In bait traps, are ants captured? Or just stay stick to the honey? It is not explained, or they can just come to eat and then go? Can this represent a source of differences? May be that smaller ants are more likely to be trapped?

L89. Figure 1 caption. State which of the sites corresponds to the picture.

The caption for Figure 2 is incomplete.

L113. foraging (without capital)

L118-120. In Methods authors said they used GLMMs, however in the Results appears as if they fitted GLMs, please clarify. Actually, it is not clear why authors fit GLMMs if they only included two fixed factors (type of trap and site). The expression “quasi-model” sounds awkward. It would be better if authors specify exactly which distribution they used for the models. Besides, this information must appear in Statistical treatment of data section, not in Results.

L120. p < 0.001 (three decimals is usually enough)

L186-187. This result is interesting indeed. Why do you consider it happens at this scale as well? Please discuss. May be small-scale conditions can have a higher relevance to this pattern? (e.g. soil, degradation, etc).

Author Response

Reviewer 1

R1: This is an interesting study but Methods and Discussion require more detail and work. Despite similar patterns have been observed in other studies, there are several issues that deserve attention and discussion. The use of mathematical tools (iNEXT package) aimed to sort out possible differences in sampling coverage is correct, but the sampling design with such huge differences in sampling effort between treatments is the main flaw of this study.

This issue was taken into consideration by computing the sampling coverage of both methods. As they both reached very high coverages (close to 100%: Fig. 4) their capture ability was saturated and therefore comparable regardless of the effort of sampling (L. 131-132).

R1: Sampling design. Please write explicitly how many traps of both types were used. Also, in this section it appears as if they collected several samples per trap but this information is missing. In the results it appears (or one has to guess) as if the number of samples is similar to the number of traps. So, did they polled all samples for each trap? Why? Or how did authors statistically treat the repeated measurements? This part needs to be much better explained, otherwise the results lack of support.

The number of bait traps was clarified to be equal and paired with the number of pitfall traps (L. 90-91). The pooling of data and its justification was added (L. 89-91)

R1: Authors should also cite for the iNEXT package: Chao, A., Ma, K. H., and Hsieh, T. C. (2016) iNEXT (iNterpolation and EXTrapolation) Online: Software for Interpolation and Extrapolation of Species Diversity. Program and User’s Guide published at http://chao.stat.nthu.edu.tw/wordpress/software_download/.

This reference was added (L.116 & 256-258)

L20. Traps

Corrected accordingly (L.20)

L83-87. In bait traps, are ants captured? Or just stay stick to the honey? It is not explained, or they can just come to eat and then go? Can this represent a source of differences? May be that smaller ants are more likely to be trapped?

Clarification on ants capture was made (L. 94-97). This methods do not rely on the ants getting stuck and therefor is not biased by ants escape capability.

L89. Figure 1 caption. State which of the sites corresponds to the picture.

Corrected accordingly (L.99).

R1: The caption for Figure 2 is incomplete.

Completed accordingly (L.100).

L113. foraging (without capital)

Corrected accordingly (L.124).

L118-120. In Methods authors said they used GLMMs, however in the Results appears as if they fitted GLMs, please clarify. Actually, it is not clear why authors fit GLMMs if they only included two fixed factors (type of trap and site).

Full model were tested and reduces to the two remaining significant factor. This was clarified (L.122)

The expression “quasi-model” sounds awkward. It would be better if authors specify exactly which distribution they used for the models. Besides, this information must appear in Statistical treatment of data section, not in Results.

The distribution information was moved to the statistical treatment of data section and was corrected for clarity (L. 117-121)

L120. p < 0.001 (three decimals is usually enough)

Corrected accordingly (L. 133)

L186-187. This result is interesting indeed. Why do you consider it happens at this scale as well? Please discuss. May be small-scale conditions can have a higher relevance to this pattern? (e.g. soil, degradation, etc).

Restoration or other smaller scale processes were not within our objectives. To properly answer this question one would need many field consideration and/or measurement that fall out of the boundaries of our study.

Reviewer 2 Report

I have only detected a minor, minor spelling error. At the top of the table in Appendix A the word stratum should be Stratum. In addition, at the bottom of figure 2 it should include "bait trap and pitfall trap".The rest of the modifications have been made conveniently.

Author Response

Reviewer 2

I have only detected a minor, minor spelling error. At the top of the table in Appendix A the word stratum should be Stratum. In addition, at the bottom of figure 2 it should include "bait trap and pitfall trap". The rest of the modifications have been made conveniently.

This was corrected accordingly (L. 100 & 228)

Reviewer 3 Report

The manuscript of Hacala et al. explores the efficiency of pitfalls and bait traps to capture the species richness and functional diversity of ant communities in heathlands along a north-south gradient at a small spatial scale. The authors used data sampled in three sites by both sampling approaches, albeit with different sampling efforts. Yet, they acknowledge the pitfalls’ higher sampling effort compared to bait traps’ effort and report adequate sampling coverage. They found higher species richness and functional diversity by the pitfall sampling approach than by the bait trapping sampling approach, attributing these differences in varying circadian activity, competitive ability, activity, and commonness of different ant species. Furthermore, they detected a north-south diversity gradient at this finer spatial scale.  Overall, I found the manuscript easy to follow in terms of structure, but I have some comments (some major and some minor).

Specific comments

-Lines 13-24: I think the “Simple Summary” should be re-written in a more concise way, while also language editing is required.

- Line 17: Do the authors mean species richness by “species counts”?

- Line 30: Both observed and estimated species richness

- Introduction: The authors should provide a brief description of the two sampling approaches, along with their advantages and shortcomings.

- Lines 46-47: The authors should re-phrase this part “and the level of knowledge in sampling methods overall follows this pattern”

- Line 49: biases related to

- Line 53: I am not sure I understand this. Do the authors mean that bait trapping has been reported as equally effective to pitfalls in capturing ant diversity?

- Line 62: with bait trapping

- Lines 60-62: The authors should justify further this expectation. Given the characteristics of each sampling approach, I would expect higher species richness in pitfalls than bait traps, and I think this expectation is also supported by the relevant literature. Is the open habitat that could make the difference? However, the pitfall approach is the main sampling approach used for sampling ant assemblages in open habitats.

- Line 75: What is “infoclimat.fr”?

- Lines 78-79: I was kind of puzzled here. Does each site include all three degradation states or one site per degradation state?  I assume the latter.

- Lines 101-102: Do the authors mean functional richness by functional diversity? Rao’s quadratic entropy and CWM are also indices of functional diversity. The authors should specify the index used here, in the remaining text, and also in the Figures.

- Lines 108-110: The authors used a Poisson GLMM to explore the differences between the two sampling approaches. Indeed, Poisson error distribution is used for count data, but functional richness and Rao's quadratic entropy are not count data. Therefore, the authors should explain why they used this error distribution in this case. Where do I see the results of these models?

- Lines 117-118: The authors should also report that there were several cases where pitfall captured a species, while bat traps did not.

- Lines 118-119: Did the authors use a quasi-poisson error distribution here?

- Line 120 and hereafter: Is this the result of the GLMM model?

-  Line 124: Table

- Table, Only bait/shared/only pitfall: Perhaps a Venn diagram provided as an inlet in the table would facilitate the reader

- Figure 3: Here statistically significant differences are depicted by asterisks, while in Figure 5 are depicted by letters.

- Line 129-130: I believe the result that sampling coverage reached asymptotes should be presented first and then the comparison of the diversity captured by the two sampling approaches.

- Line 130: sufficient or adequate sampling intensity

- Line 131: Which estimator was used for the estimated species richness?

- Line 131: Figs 4b and 4c

- Caption of figure 4: A more accurate figure caption is required. This stands for all figure captions.

- Line 144: taxonomic and functional diversity metrics

- Line 145: lower functional diversity

-  Lines 155-156: The authors should re-phrase.

- Lines 160-161: I assume similar trends in capturing only a fraction of target taxon’s diversity. Wang et al. (2001) found that pitfalls were superior to bait traps for leaf litter ants using data from forests. This can be used to justify the first expectation, i.e. open habitats vs. forests. As a matter of fact, all the arguments for authors’ findings suggest that their original expectation should be re-formulated, i.e. pitfalls would capture higher species richness than bait trapping and that they are testing for the validity of this hypothesis in open habitats.  

- Lines 191-198: This part should be re-written, it  is difficult to follow. :

- Lines 184-186: But the authors state in Lines 59-60 that the degradation gradient will not affect the observed patterns due to the small spatial scale.

- Line 193: multiple

- Line 194: I did not find Mahon et al. in the reference list.

I think that these issues need to be addressed.

I hope the authors find the aforementioned comments and suggestions helpful to further improve their manuscript.

Author Response

Reviewer 3

-Lines 13-24: I think the “Simple Summary” should be re-written in a more concise way, while also language editing is required.

A simplification effort was undertake to ease the reading of the “simple summary” (L. 13-22)

- Line 17: Do the authors mean species richness by “species counts”?

Yes

- Line 30: Both observed and estimated species richness

The missing word “richness” was added (L.30)

- Introduction: The authors should provide a brief description of the two sampling approaches, along with their advantages and shortcomings.

Information was added in that regard (L. 47-52)

- Lines 46-47: The authors should re-phrase this part “and the level of knowledge in sampling methods overall follows this pattern”

Rephrased accordingly (L 46-47)

- Line 49: biases related to

Corrected accordingly (L.54)

- Line 53: I am not sure I understand this. Do the authors mean that bait trapping has been reported as equally effective to pitfalls in capturing ant diversity?

We were highlighting the comparison between methods but did not mean to describe the results of those comparison here. The results were consistent with ours. The sentences was change to improve clarity (L. 59-60)

- Line 62: with bait trapping

Corrected accordingly (L.68)

- Lines 60-62: The authors should justify further this expectation. Given the characteristics of each sampling approach, I would expect higher species richness in pitfalls than bait traps, and I think this expectation is also supported by the relevant literature. Is the open habitat that could make the difference? However, the pitfall approach is the main sampling approach used for sampling ant assemblages in open habitats.

The expectation was changed according to other comments and to fit literature (L. 65-68).

- Line 75: What is “infoclimat.fr”?

An online meteorological database with a regional scale of precision for French territories which the meteorological data were acquired from. Precision was added to clarify this point (L. 81).

- Lines 78-79: I was kind of puzzled here. Does each site include all three degradation states or one site per degradation state?  I assume the latter.

Each site include the three degradation states. Clarification was added (L. 85)

- Lines 101-102: Do the authors mean functional richness by functional diversity? Rao’s quadratic entropy and CWM are also indices of functional diversity. The authors should specify the index used here, in the remaining text, and also in the Figures.

Functional richness was meant and corrected throughout the manuscript when functional diversity was used.

- Lines 108-110: The authors used a Poisson GLMM to explore the differences between the two sampling approaches. Indeed, Poisson error distribution is used for count data, but functional richness and Rao's quadratic entropy are not count data. Therefore, the authors should explain why they used this error distribution in this case. Where do I see the results of these models?

This distribution problem was addressed by switching to Gaussian distribution for functional richness and Rao’s quadratic entropy. The tendency remain the same and the manuscript was update both in method (L. 118) and in the results (L. 154 & 158).

- Lines 117-118: The authors should also report that there were several cases where pitfall captured a species, while bat traps did not.

This information was highlighted (L. 129-130)

- Lines 118-119: Did the authors use a quasi-poisson error distribution here?

yes

- Line 120 and hereafter: Is this the result of the GLMM model?

yes

-  Line 124: Table

Corrected accordingly (L.137)

- Table, Only bait/shared/only pitfall: Perhaps a Venn diagram provided as an inlet in the table would facilitate the reader

A Venn diagram was added to table 1 (L.139), thanks for this suggestion.

- Figure 3: Here statistically significant differences are depicted by asterisks, while in Figure 5 are depicted by letters.

This was corrected on fig 4 by homogenizing with letters (L.147)

- Line 129-130: I believe the result that sampling coverage reached asymptotes should be presented first and then the comparison of the diversity captured by the two sampling approaches.

Figure order between fig. 3 and 4 was inverted

- Line 130: sufficient or adequate sampling intensity

Corrected accordingly (L.132)

- Line 131: Which estimator was used for the estimated species richness?

Chao’s estimator implemented within iNext

- Line 131: Figs 4b and 4c

Changed accordingly (L.141)

- Caption of figure 4: A more accurate figure caption is required. This stands for all figure captions.

Figure 4’s capitation has been improved with a more thorough description.

- Line 144: taxonomic and functional diversity metrics

Added accordingly (L.159).

- Line 145: lower functional diversity

Corrected accordingly (L.160)

-  Lines 155-156: The authors should re-phrase.

The phrase was re-written in a more understandable way (L.169-171).

- Lines 160-161: I assume similar trends in capturing only a fraction of target taxon’s diversity. Wang et al. (2001) found that pitfalls were superior to bait traps for leaf litter ants using data from forests. This can be used to justify the first expectation, i.e. open habitats vs. forests. As a matter of fact, all the arguments for authors’ findings suggest that their original expectation should be re-formulated, i.e. pitfalls would capture higher species richness than bait trapping and that they are testing for the validity of this hypothesis in open habitats.  

The initial assumption was due changed to match pitfall being superior (L. 65-68) and the discussion was corrected accordingly as well (L.168 -172).

- Lines 191-198: This part should be re-written, it is difficult to follow. :

The whole paragraph was reworked to ease the reading (L. 206-214)

- Lines 184-186: But the authors state in Lines 59-60 that the degradation gradient will not affect the observed patterns due to the small spatial scale.

The small spatial scale was invoked to justify not to test for restoration effect in this study. This choice was made to remain coherent with our objectives and scale but never meant to prove degradation to have no possible effects.

- Line 193: multiple

“Multiplying sampling methods is costly” was kept as we meant that the using several method at once was costly.

- Line 194: I did not find Mahon et al. in the reference list.

The reference was added to the list (L. 307-309)

Round 2

Reviewer 1 Report

The paper provides relevant methodological and biological information. The suggestions were answered or included properly, except for the last point (L186-187. Why do you consider it happens at this scale as well? Please discuss). I understand there are many issues that can scape to our studies. However, this particular issue (North-South gradient) was one of the initial hypotheses and deserves a more comprehensive discussion. 

Author Response

The paper provides relevant methodological and biological information. The suggestions were answered or included properly, except for the last point (L186-187. Why do you consider it happens at this scale as well? Please discuss). I understand there are many issues that can scape to our studies. However, this particular issue (North-South gradient) was one of the initial hypotheses and deserves a more comprehensive discussion. 

Explication were added (L.209-211).

Reviewer 3 Report

After reading the changes made in the manuscript, I found that the authors have taken care to amend their manuscript in accordance with suggestions. However, further effort is required as some parts need to be revised more carefully.

Specific comments (lines refer to the “track-change” manuscript):

- Line 47-48: “and the level of knowledge … for tropical regions” this part does not make sense.

- Lines 47-60: The authors after my suggestion, included a brief description of the two sampling methods (Lines 47-52), but this description should be more smoothly incorporated into the text.

 - Lines 47-48: “The two methods…bait traps” re-phrase e.g. Two common sampling methods for ant assemblages are pitfall and bait traps.

- Line 49: Do the authors mean “sensitive to” ?

- Line 51: What is mFay?

- Line 69: The authors should keep the term functional diversity here.

- Line 81: The authors should provide the reference for this page: Infoclimat. Pourcentage d' Humidité relative. Available …., Accessed ….

- Line 85: per site

- Line 90: trap

- Line 91: consisted by

- Line 94: all the ants present on the trap

 - Line 95: remained active for 12 weeks

-  Lines 95-97: For each …., to assess local assemblages

- Lines 120-122: The authors should re-phrase. Furthermore, the authors should give the full name of the model used and then the abbreviation.

- Line 125: What is decremential fitting?

- Line 148-150: I believe that in the revised ms this is Figure 3b and c.

- Lines 168-171: The authors should re-phrase

- Lines 180-181: The authors should re-phrase

- Lines 198-199: of the two sampling methods is that bait traps captured species that are highly abundant and active

- Lines 209-210: The authors should re-phrase

- Line 219: the results are consistent

Author Response

After reading the changes made in the manuscript, I found that the authors have taken care to amend their manuscript in accordance with suggestions. However, further effort is required as some parts need to be revised more carefully.

Specific comments (lines refer to the “track-change” manuscript):

- Line 47-48: “and the level of knowledge … for tropical regions” this part does not make sense.

This was rephrased in order to clarify what we intended by it (L. 46).

- Lines 47-60: The authors after my suggestion, included a brief description of the two sampling methods (Lines 47-52), but this description should be more smoothly incorporated into the text.

The description is now accompanied with a contextualization that smoothen the reading. (L. 47-49, 49-50)

 - Lines 47-48: “The two methods…bait traps” re-phrase e.g. Two common sampling methods for ant assemblages are pitfall and bait traps.

The change have been applied accordingly (L. 49)

- Line 49: Do the authors mean “sensitive to” ?

Yes, and it was changed accordingly (L.52)

- Line 51: What is mFay?

It’s a typo which was corrected (L. 50).

- Line 69: The authors should keep the term functional diversity here.

This was corrected accordingly (L.71).

- Line 81: The authors should provide the reference for this page: Infoclimat. Pourcentage d' Humidité relative. Available …., Accessed ….

This as update accordingly (L. 84)

- Line 85: per site

Corrected accordingly (L. 88).

- Line 90: trap

Corrected accordingly (L. 93)

- Line 91: consisted by

This was corrected but “consisted by” sounded wrong and some research on what should follow “consisted” in our context lead to choose “of” for it fits with the material nature of the description (L.94).

- Line 94: all the ants present on the trap

Corrected accordingly (L.97).

 - Line 95: remained active for 12 weeks

Corrected accordingly (L. 98).

-  Lines 95-97: For each …., to assess local assemblages

Corrected accordingly (L.100).

- Lines 120-122: The authors should re-phrase. Furthermore, the authors should give the full name of the model used and then the abbreviation.

Corrected accordingly (L. 121-122, 125).

- Line 125: What is decremential fitting?

We tested full model that comprise all variable and their interaction and once no interaction were detected we removed them and had a simpler model. Then we removed non-significant variables, step by step simplifying the model, decrementially, up to its simplest form where only significant variables remains. Although this procedure is quite common in statistics, we can explain it in the ms if needed.

- Line 148-150: I believe that in the revised ms this is Figure 3b and c.

Corrected accordingly (L.144).

- Lines 168-171: The authors should re-phrase

The passage was re-phrase for clarity (L.171-173).

- Lines 180-181: The authors should re-phrase

The passage was re-phrase for clarity (L.182-183).

- Lines 198-199: of the two sampling methods is that bait traps captured species that are highly abundant and active

Both methods captured those species, the differences being that the other ants were capture by pitfall only.

- Lines 209-210: The authors should re-phrase

Rephrase for clarity (L. 216).

- Line 219: the results are consistent

We must apologies for we don’t understand this comment that appear to have been truncated.

This manuscript is a resubmission of an earlier submission. The following is a list of the peer review reports and author responses from that submission.

Round 1

Reviewer 1 Report

Hacala et al investigated the effectiveness and complementarity of pitfall trapping and baiting at three temperate habitats with various levels of degradation. The authors found that baiting captured smaller number of species and no species was unique to the baiting compared with those collected by pitfall trapping. The authors concluded that, in their study system, baiting method maybe redundant.

This study unfortunately misses other, more commonly utilised sampling methods, such as litter extraction, arboreal baiting and manual hand collection, that targets different microhabitats of ants that pitfall trapping often does not capture. As discussed by the authors, it is not surprising that baiting deployed at the same microhabitat as pitfall traps did not work as a complementary method, as this method primarily targets the same set of ant species. Although the authors only focused on species and functional richness of these two methods and concluded that baiting may be redundant, bating is valuable as this shows another aspect of ant ecology related to competition.

If the aim of this study is to find ways to effectively sample ants within a given habitat, other methods (e.g., litter extraction) should have been included. If the aim of this study was testing the effectiveness of baiting in ant ecological studies, then its usefulness to assess competitive strengths of an ant community within a given habitat should have been assessed.

Other minor comments are provided as follows:

 L89: The number of pitfall traps (71) does not add up according to the description: 3 sites x 2 plots x 4 traps per plot (minus one trap removed due to small habitat area).

L109: It is not clear as to what was used as independent statistical units for analysis (plots?). Please also provide degrees of freedom so that the readers can see the number of statistical units.

L115: Rarefaction curves provided in your study shows gamma diversity, not alpha diversity, which is average number of species across samples.

L122-123: The authors claim that site influence is important as the diversity of ants increase with increasing temperature, but no environmental information was provided for the sites. At least basic information such as canopy coverage, average temperature and rainfall should be provided if the authors believe that temperature and latitudinal gradient (L210-214) are important.

L128-130: It is not clear why the author conducted separate analysis to test the effect of site differences using non-parametric tests. The site effects are already included in the GLMMs described 121-124.

Figure 5: Are the graphs based on the pitfall tapped ants? Please clarify in the caption.

L160: “Both methods confounded”? What do you mean? Did you mean to say “Both methods confirmed”? If both methods showed the same patterns in assemblage composition, then the authors could suggest that baiting is also a good method to see the differences in ant species composition among different habitats, and baiting is generally easier to run than pitfall trapping. The authors should consider sampling effort (time and money) required to carry out pitfall trapping and baiting and then consider which methods are more effective for this purpose.

Reviewer 2 Report

In the manuscript “Relative efficiency of pitfall vs. bait trapping for capturing taxonomic and functional diversities of ant assemblages in temperate heathlands” authors aim to evaluate the efficiency of pitfall and baited traps to assess ant communities. The hypothesis is sound and it seems researchers made an important quantity of field and laboratory work.

What worries me more is that the sampling effort with both types of trap seems radically different. For my calculations (although this should have been clearly stated by authors): Pitfall traps:  3 months (March-June)* 30 days/month * 24 hours /day = 2160 hours/trap. Baited traps: 5 times * 2 hours/time= 10 hours/trap. If this is true, the experiment has an important problem of design. Since the type of trap with higher effectiveness was the one with the higher sampling effort it is possible that the results are not due to the type of trap, but a flaw in sampling effort.

Other issues:

The work deals with a lot of factors that are presented without a clear link. Most importantly, it is not clear the relevance of including them to answer the main question.

The methods need much more detail and accuracy. For instance, the sample size (alias "N") is a mystery to be solved by the lector.

Please, review and correct numerous typos scattered throughout the text (points, numbers, capitals, others) before submitting a manuscript to a journal.

L171. Chaos has written many chapters in science, but I wonder you mean Chao.

The language needs to be considerably improved. Please let an editorial service to make corrections.

Reviewer 3 Report

  • In figure number 4 each graph must be specified with its corresponding letter.
  • In section 2.2. Sampling design, it would be convenient to include the image of pitfall.
  • In line 167 it would be convenient to change "with groups of letters" to "with different letters"
  • On lines 86-88. The pitfall sampling design is not clear enough.
  • The table in Appendix A presents a lot of information and is very compressed. It would be convenient to use acronyms or initial letters and attach a small legend at the end of it.
  • Finally, I have trouble understanding the table in Appendix B correctly. It would be convenient to divide each trait with its description because they are mixed.